# NLRP3 Inflammasome in Acute and Chronic Liver Diseases

**DOI:** 10.3390/ijms25084537

**Published:** 2024-04-20

**Authors:** Katia Sayaf, Sara Battistella, Francesco Paolo Russo

**Affiliations:** 1Department of Surgery, Oncology and Gastroenterology, University of Padova, 35128 Padua, Italy; katia.sayaf@phd.unipd.it (K.S.); sarabattistella93@gmail.com (S.B.); 2Gastroenterology and Multivisceral Transplant Unit, Padua University Hospital, 35128 Padua, Italy

**Keywords:** acute liver injury, chronic liver disease, NLRP3 inflammasome, liver transplantation

## Abstract

NLRP3 (NOD-, LRR-, and pyrin domain-containing protein 3) is an intracellular complex that upon external stimuli or contact with specific ligands, recruits other components, forming the NLRP3 inflammasome. The NLRP3 inflammasome mainly mediates pyroptosis, a highly inflammatory mode of regulated cell death, as well as IL-18 and IL-1β production. Acute and chronic liver diseases are characterized by a massive influx of pro-inflammatory stimuli enriched in reactive oxygen species (ROS) and damage-associated molecular patterns (DAMPs) that promote the assemblage and activation of the NLRP3 inflammasome. As the major cause of inflammatory cytokine storm, the NLRP3 inflammasome exacerbates liver diseases, even though it might exert protective effects in regards to hepatitis C and B virus infection (HCV and HBV). Here, we summarize the current knowledge concerning NLRP3 inflammasome function in both acute and chronic liver disease and in the post liver transplant setting, focusing on the molecular mechanisms involved in NLRP3 activity.

## 1. Introduction

Inflammasomes comprise a large family of intracellular receptors formed by multiple and complex proteins that are all involved in the activation of proinflammatory cytokines such as interleukin-1β (IL-1β), interleukin-18 (IL-18), and tumor growth factor-β (TGF-β) [1]. Inflammasomes are widely expressed in both immune and non-immune cells, thus acting as key factors in the onset of many autoimmune and non-autoimmune-based liver diseases [2,3].

Within the hepatic environment, inflammasomes are abundant in the cytoplasm of monocytes, B cells, and T cells, while the non-immune cells include hepatic stellate cells (HSCs) and their activated form, i.e., fibroblasts and myofibroblast, together with parenchymal cells [3,4]. As a consequence of cellular damage, the resulting pathogen-associated molecular patterns (PAMPs), damage-associated molecular patterns (DAMPs), and reactive oxygen species (ROS) are identified by inflammasome proteins recognized through pathogen recognition receptors (PPRs) [3]. These proteins, known as sensor proteins, include nucleotide-binding domain-like receptors (NLRs) such as NLRP1 and NLRP3, lacking the melanoma 2-like receptors (ALRs) that upon external stimuli or contact with specific ligands, recruit the adaptor protein ASC (apoptosis-associated speck-like protein), inherently containing a caspase-activation domain (CARD) [5,6]. The newly formed complex recruits the effector protein procaspase-1 by CARD-CARD interactions, leading to its cleavage and autoactivation to caspase-1, which in turn cleaves pro-IL-18 and pro-IL-1β into their mature forms [6]. The autoactivation of caspase might also be mediated by its direct binding to lipopolysaccharide (LPS) on Gram-negative bacteria, resulting in the cleavage of the pore-forming protein gasdermin-D (GSDMD) and cell death [3,5,7].

### 1.1. Inflammasome Activation: The Priming Step

In regard to NLRP3 inflammasome activation, a priming phase is required. During the priming phase, pro-inflammatory stimuli, i.e., DAMPs, trigger PPRs, such as Toll-like receptors (TLRs, in particular TLR4) and nucleotide-binding oligomerization domain-containing protein-2 (NOD2), leading to the activation of NF-kB and gene transcription. It has been discovered that this process is also activated by cytokines such as tumor necrosis factor α (TNF-α) and IL-1β [3,5,6]. The first function of the priming phase is to upregulate the components of the inflammasome, including NLRP3, pro-caspase-1, and caspase-1, together with pro-IL-1β, while the second function consists of the induction of post-translational modifications (PMTs) of NLRP3, such as ubiquitylation and phosphorylation, that stabilize the protein in an inactivated state, preparing it for activation upon stimulation [6,8,9].

### 1.2. Inflammasome Activation: The Activation Step

During the activation step, the activator (ASC) is recognized by NLPR3 and recruits pro-caspase-1 to complete the formation of the inflammasome complex that is now capable of producing mature IL-1β and IL-18 [9]. NLRP3 activation includes multiple upstream complexes, which will be briefly described in this review. These include the efflux of potassium (K^+^) and chloride ions (Cl^−^), the influx of calcium ions (Ca^2+^), and mitochondrial dysfunction.

Based on the K^+^ hypothesis, ATP recognizes its receptor (P2X7 purinergic receptor, P2X7R) on the surface of the cell membrane, thus opening the K^+^ channels and leading to an outflow of K^+^ ions. In the meantime, PAMPs enter the cell and activate the NLRP3 inflammasome [3]. Low extracellular levels of Cl^−^ assists in ATP-induced IL-1β secretion, whereas blocking the Cl^−^ channel, as well as high extracellular levels of this ion, cause NLRP3 inhibition, thus suggesting that Cl^−^ efflux contributes to NLRP3 activation [9]. Phospholipase C (PLC) is responsible for the hydrolyzation of phosphatidylinositol-4,5 diphosphate to inositol trisphosphate (InsP3). The latter, upon binding to its receptor on the endoplasmic reticulum, allows for the efflux of Ca^2+^, leading to an increase in this ion in the intracellular space. Calcium-sensing receptors (CASRs) recognize Ca^2+^ and activate NLRP3 inflammasome [3]. Mitochondrial dysfunction is responsible for an increase in the mitochondrial ROS (mtROS) into the cytosol, which in turn leads to NLRP3 activation. Mitophagy, a process that deletes dysfunctional mitochondria, might be a useful target for decreasing the production of ROS and regulating inflammasome activation [9] (Figure 1). Thus, this review aims to discuss and analyze the role of the frequently studied inflammasome NLRP3 in acute and chronic liver diseases, as well as the manner in which NLRP3 might orchestrate complications after liver transplantation (LT).

## 2. The Role of the NLRP3 Inflammasome in Acute Liver Injury (ALI)

Different etiologies, such as viral hepatitis, drugs, ischemia-reperfusion damage, and autoimmune and alcohol-associated hepatitis, have been identified as causative factors of acute liver injury (ALI) [10]. ALI can rapidly progress to acute liver failure (ALF), and, at that point, LT is the only curative option [11]. Several studies on humans and mice reported that the NLRP3 inflammasome plays a key role in the pathogenesis of both alcohol- associated hepatitis and drug-induced liver injury (Table 1).

### 2.1. Drug-Induced Liver Injury (DILI)

Drug-induced liver injury (DILI) is one of the most common causes of ALI and ALF in Western countries, and acetaminophen (APAP) is the most frequent drug responsible for DILI [12,13]. At therapeutic doses, APAP is principally transformed into non-toxic molecules that are excreted with the urine, and only a small part is metabolized in N-acetyl-p-benzoquinone (NAPQI). NAPQI is a cytotoxic compound that is detoxified through the binding with glutathione (GSH) in the liver. However, the excessive assumption of APAP results in GSH depletion. The excessive accumulation of NAPQI causes mitochondrial dysfunction, the production of reactive oxygen species, and the release of apoptosis-inducing factor (AIF). The AIF translocates into the nucleus and causes hepatocyte necrosis which, in turn, determines the release of DAMPs, recognized by the TLR of the macrophages, resulting in NLRP3 inflammasome activation [13].

Recently, it was found that pyroptosis occurs in both hepatocytes and Kupffer cells via GSDMD activation in APAP-induced liver injury. To confirm this fact, Yuan et al. demonstrated that NLRP3 activation post-APAP administration exhibits the same kinetics as pyroptosis-related proteins, cleaved caspase-1, GSDMD, and cleaved IL-1β. In the same study, the pharmacological inhibition of the NLRP3/GSDMD pathway by MCC950 yields the same result observed in Nlrp3^Δhep^ (hepatocyte specific Nlrp3 deficiency). Directly targeting the NLRP3 inflammasome might be a good therapeutic strategy for inhibiting hepatocyte pyroptosis and alleviating the inflammatory response [14].

On the other hand, targeting the upstream signals responsible for NLRP3 inflammasome activation might also be a valid strategy. In fact, recent discoveries have underlined the importance of phytotherapeutic components in reducing inflammation by targeting the NLRP3 inflammasome. In the context of APAP-induced hepatotoxicity, research has demonstrated that Kaempferol, a flavonoid molecule, can inhibit the activation of the NLRP3 inflammasome by protecting the liver from inflammation and consequent apoptosis [15] (Table 1). To test the theory that phytotherapeuticly derived molecules might be an important source of components to counteract APAP-induced hepatocellular damage, Elshal et al. studied the effects of diacerein, an anthraquinone, in mice. They found lower rates of oxidative stress, necrosis, and hepatic inflammation in diacerein-treated mice as a result of reduced NLRP3 activity driven by a downregulated NF-kB pathway [16]. Other recent studies focused on the role of Necrostatin-1 (Nec-1), an inhibitor of receptor-interacting serine-threonine kinase (RIPK1), which in turn, plays an important role in the activation of the NLRP3 inflammasome. Therefore, the inhibition of RIPK1 is associated with lower APAP-induced liver damage [17].

Wang et al. studied the effect of Peroxideroxin 3 (PRX3), a member of the thiol peroxidase family, in the regulation of NLRP3-mediated pyroptosis in APAP-mediated hepatotoxicity. As a scavenger of peroxidase in cells, PRX3 was found to protect against APAP-induced pyroptosis by targeting mitochondrial ROS and inhibiting the NLPR3 inflammasome activation [18,19] (Table 1). Nevertheless, other studies did not provide evidence supporting the involvement of the NLRP-3 inflammasome in exacerbating APAP-induced liver injury [20,21]. Consequently, additional research is required to ascertain its role and its potential therapeutic implications.

### 2.2. Alcohol-Associated Hepatitis

Alcohol-associated hepatitis is characterized by the onset of acute jaundice, coagulopathy, and liver test alteration in patients with excessive alcohol intake [22]. The factors responsible for alcohol-associated hepatitis are multiple and include environmental, genetic, and epigenetic factors [23]. The liver is the organ most affected by alcohol abuse, since it metabolizes alcohol in acetaldehyde, which in turn forms DNA adducts that trigger inflammatory responses such as lipid peroxidization, innate immune responses, and mitochondria damage [23]. The pathogenesis of alcohol associated hepatitis is indeed multifactorial and is the result of an interplay between ethanol metabolism, inflammation, and bacterial translocation. Excessive and chronic alcohol consumption leads to a disrupted hepatic lipid metabolism, as well as an impaired intestinal barrier integrity that allows for the translocation of endotoxins (lipopolysaccharide) to the liver [24]. The bond between the latter and liposaccharide-binding proteins results in the formation of a complex that induces Kupffer cell (KC) activation, perhaps via the NLRP3 inflammasome, and TNF production by the Th1 compartment. By binding to TNFR1, TNF is responsible for hepatocyte necrosis and apoptosis [25]. Considering their additional role as TNF inhibitors, steroids are the only therapeutical approach, even though some patients with severe alcoholic hepatitis (sAH) do not respond to corticosteroid therapy and rapidly progresses to ALF [26].

Excessive ethanol consumption leads to a plethora of metabolic dysfunctions. Ethanol itself increases xanthine oxidase activity, leading to the production of superoxide. Chronic alcohol intake induces the cytochrome P450 2E1 isoform, serving as a catalyst for the formation of ROS. Additionally, alcohol enhances aldehyde oxidase activity, contributing to the generation of oxyradicals. Lastly, structural alterations in the mitochondria, one of the earliest effects of ethanol consumption, are implicated in intracellular ROS production [21]. Moreover, excessive amounts of acetaldehyde activate the KCs to release ROS and other chemokines that recruit neutrophils in the liver, which are responsible for hepatocyte killing [27]. ROS also promote the activation of the NLRP3 inflammasome via the NF-kB pathway, leading to the production of pro-inflammatory cytokines, e.g., TNF, IL-1β, and IL-8, which are upregulated in patients with alcohol-related liver damage [28] (Table 1). Upon binding to its receptor on the KCs, IL-1β acts as a mediator of liver inflammation, since it sustains the activation of resident macrophages and contributes to hepatitis and fibrosis [29,30].

Multiple trials targeting IL-1β have been proposed. The DASH trial indicates that the combination of the interleukin-1β receptor antagonists anakinra, pentoxifylline, and zinc do not enhance survival compared to the use of corticosteroids in patients with sAH. In the AlcHepNet trial, which compared the current standard of care, prednisone, with anakinra plus zinc, an interim analysis prompted early termination because patients treated with prednisone showed superior 90-day overall survival and transplant-free survival rates, along with a lower incidence of acute kidney injury compared to those treated with anakinra plus zinc (ClinicalTrials.gov ID NCT04072822, Version 6: 6 January 2022, accessed on 8 January 2024).

By targeting the redox reactions upstream of NLRP3, Liu et al. showed that quercetin, a plant-derived polyphenol, can reduce the production of ROS as a consequence of promoting the expression of heme oxygenase-1 (HO-1), an antioxidant and metabolic regulator. Interestingly, they found that quercetin alleviates acute alcohol-related liver injury (ALI) in mice by increasing the expression of HO-1, which in turn inhibits the activation of the NLRP3 inflammasome [31]. Hence, quercetin could represent a promising compound for mitigating ethanol-induced damage, although further research is necessary to comprehensively elucidate its effects and ascertain the safety of this approach.

Metadoxine, another antioxidant drug, was found to block the general secretion of TNF and has been associated with accelerated alcohol elimination, along with recovery from alcohol injury, in a small randomized-controlled clinical trial [32,33] (Table 1). Furthermore, Metadoxine was reported to increase the steroid response in patients with alcohol-associated hepatitis, improving 30- and 90-day survival [34].

The FXR agonist obeticholic acid (OCA) was proposed as a potential therapy for patients with alcohol-associated hepatitis due to its anti-oxidative properties. However, after a warning from the FDA about safety issues regarding higher dosages of OCA in patients with severe acute hepatitis, the phase 2 clinical trial (NCT02039219) was halted. Metadoxine and OCA might be a valid pharmacological treatments for alcohol-associated hepatitis, since they are antioxidant drugs that may target NLRP3 inflammasome activation by reducing ROS levels. However, the involvement of these two drugs in the inhibition of the NLRP3 inflammasome should be further investigated.

**Table 1 ijms-25-04537-t001:** Main studies investigating the involvement of the NLRP3 inflammasome in acute liver diseases. Abbreviations: DILI: drug-induced liver injury; NF-kB: nuclear factor kappa B; PRX3: peroxideroxin 3; HO-1: heme oxygenase-1; OCA: obeticholic acid; ROS: reactive oxygen species; TLR4: Toll-like receptor 4; NEC-1: necrostatin-1; RIPK-1: receptor-interacting serine/threonine-protein kinase 1.

Type of ALF	Effect	Sex and Species	Reference
	Kaempferol inhibits NLRP3 inflammasome activation.	Male C57BL/6 mice	Research paper [15]
DILI	Diacerein reduces the activity of the NLRP3 inflammasome by downregulating NF-kB.	Adult Balb/c mice	Research paper [16]
	PRX3 protects against APAP-induced pyroptosis by inhibiting NLRP3 inflammasome activation.	Male C57BL/6 mice	Research paper [18]
	Acetaldehyde promotes ROS production, which activates the NLRP3 inflammasome via NF-kB.		Review [28]
Alcohol-Associated Hepatitis	Quercetin reduces ROS production and increases HO-1, which in turns inhibits the activation of the NLRP3 inflammasome.	Male SPF-Wistar rats	Research paper [31]
	Metadoxine and OCA inhibit ROS production, perhaps leading to reduced NLRP3 activation.		Review [32]

## 3. NLRP3-Mediated Inflammation in Chronic Liver Diseases

Chronic liver diseases are a major public threat and are characterized by a progressive deterioration of liver function due to multiple etiologies such as viral infections, metabolic dysfunction-associated steatotic liver disease, and alcohol abuse [35,36] (Table 2).

### 3.1. Chronic Viral Hepatitis C (HCV)

Hepatitis C virus (HCV) is a hepatotropic enveloped virus that carries a single-stranded positive-sense RNA genome. This virus infects hepatocytes and is responsible for acute liver inflammation that can become chronic, leading to fibrosis, cirrhosis, and hepatocellular carcinoma (HCC). Inflammation is the first step after HCV invasion and is characterized by increased levels of NLRP3-related proteins such as IL-1β and IL-18 [1]. A study from Negash et al. found that HCV-induced inflammasome activation is triggered by the attachment of viral RNA to endosomal TLR7 in Kupffer cells, leading to IL-1β production via MyD88. Additionally, HCV invasion also leads to an increase in K^+^ efflux, thus activating the NLRP3 inflammasome and the production of IL-1 β, which is responsible for triggering other inflammatory signals [37] (Table 2).

In hepatocytes, HCV, as suggested by McRae et al., upregulates NLRP3 and caspase-1 to create the perfect environment for its replication [1] (Table 2). In addition, a recent study regarding programmed cell death has confirmed the involvement of NLRP3 inflammasome in HCV-induced pyroptosis, a pro-inflammatory process of cell death mediated by caspase-1 and GSDM-D. In this study, Wallace et al. demonstrated that pyroptosis occurs before apoptosis during HCV infection, with an involvement of NLRP3 and caspase-3, and that these processes lead to increased viral replication [38]. An augmented viral load leads to sustained inflammatory processes which are responsible for the onset of many chronic liver diseases, such as cirrhosis and ultimately, hepatocellular carcinoma (HCC) [39].

Among the main causes of HCC from HCV infection is the activation of the phosphatidylinositol (PI3K)/protein kinase B (Akt)/mammalian target rapamycin (mTOR) pathway, which can be triggered by multiple damage-associated factors such as pro-inflammatory cytokines, TLR ligands, and growth factors [40]. The HCV-mediated activation of the NLRP3 inflammasome in macrophages may play an important role in the onset of liver fibrosis, since it secretes IL-1β, which in turn regulates TGF-β1 release. Although both chemoattractant molecules are master regulators of fibrogenic processes, TGF-β1 also activates the PI3k/Akt/mTOR pathway, a crucial pathway upregulated in HCC [40,41]. Targeting NLRP3 inflammasome activation might be a valid approach to both reduce HCV-infected cell death and diminish the activation of fibrogenic processes that lead to cancer.

### 3.2. Chronic Viral Hepatitis B (HBV)

Hepatitis B virus (HBV) is an oncogenic, double-stranded, hepatotropic DNA virus which can lead to chronic hepatitis, and eventually, to the development of cirrhosis and HCC [42]. Findings from Xie et al. suggested that the protein HBx could activate the NLRP3 inflammasome in normal hepatocytes and enhance NLRP3 inflammasome-mediated inflammation and pyroptosis by upregulating the production of ROS in the mitochondria (mtROS). Thus, upon HBx-mediated infection, the activation of the NLRP3 inflammasome was confirmed by a very high secretion of NLRP3-associated proteins such as ASC, IL-1β, and IL-18 [43] (Table 2).

As a double edge swarded, the NLRP3 inflammasome might play a role in the eradication of HBV. The NLRP3 inflammasome, together with other NLRs such as NLRP12 and NLRP1, might trigger humoral immunity against HBV, since they trigger immune responses against the hepatitis B surface antigen (HBsAg) vaccine [1]. Another study that supports the positive effects of the NLRP3 inflammasome on HBV infection is provided by Yu et al., as they demonstrated that upon LPS stimulation, HBV infection blocks the induction of NLRP3, as well as IL-1β production, in Kupffer cells. The agent responsible for this inhibitory activity is the hepatitis B e-antigen (HBeAg) that suppresses IL-1β maturation by repressing the NF-kB pathway and ROS production [44].

As for HCV, persistent levels of HBV in infected cells lead to the exhaustion of the HBV-specific T cells, which is characterized by the presence of inhibitory receptors such as programmed-death protein 1 (PD1) and cytotoxic T-lymphocyte antigen 4 (CTLA4). A few studies in murine models showed that KCs mediate the exhaustion of cytotoxic T lymphocytes (CTLs) by PDL1-PD1 interaction. KCs might also support the retention of antiviral CD4+ T cells in the liver and their apoptosis by the secretion of CXCL9 [45]. To date, only the correlation between NLRP3 and CXCL9 suggests that NLRP3 has a role in mitigating CXCL9 secretion [46].

A study from Li and Juang demonstrated that PBMCs from patients with chronic HBV did not show an increased mRNA expression of NLRP3 or serum levels of IL1β when compared to those of healthy controls, suggesting that the inflammasome might not be involved in inducing chronic HBV [47]. However, the study investigated the role of the NLRP3 inflammasome in the peripheral blood, and not in liver samples. While investigating the crosstalk between innate and adaptative immunity in chronic HBV, the role of the NLRP3 inflammasome has been poorly elucidated and needs to be further unraveled.

### 3.3. MASLD and MASH

Metabolic dysfunction-associated steatotic liver disease (MASLD) is a comprehensive term encompassing all grades and stages of the disease, referring to a population in which ≥5% of hepatocytes exhibit macrovesicular steatosis. The disease spectrum might range from steatosis, a simple fat deposition in the hepatic parenchyma, to metabolic dysfunction-associated steatohepatitis (MASH), usually associated with insulin resistance and obesity [48]. This hepatic metabolic impairment is characterized by the presence of inflammation, with or without fibrosis. MASH can then progress to cirrhosis, a condition identified by bands of fibrous septa leading to the formation of cirrhotic nodules [29,49]. Increased levels of NLRP3 inflammasome components, such as caspase-1, were found in the sera of MASLD patients, and they correlated with the presence of liver damage [50]. Consistent results were also found in murine models of MASH, since hepatic mRNA levels of caspase-1 and sera levels of IL-1β were abundant, while the suppression of NLRP3 was responsible for a decreased hepatic expression of this protein, along with a lowering of hepatic and circulating IL-1β, IL-6, and CCL2 [51,52] (Table 2).

By blocking NLRP3, Mridha et al. found that the pharmacological drug MCC950 did not improve steatosis or cholesterol deposition, while it abrogated the pro-inflammatory effects typical of steatohepatitis [51]. Controversially, the loss of NLRP3 in a murine model of MASLD ameliorated hepatic steatosis and protected macrophage recruitment [4] (Table 2).

Wu et al. discovered that rhubarb-free anthraquinones (RFAs) might be a potential therapeutic agent for MASLD, since they directly inhibited the assembly of NLRP3 in vitro, but not NLRP3 protein expression, and improved histopathological inflammation in MASLD in mice fed a methionine and choline deficient diet [53].

Among the NLRP3 inflammasome activators and initiators of MASLD are extracellular adenosine triphosphate (ATP), mtROS, and mitophagy. Upon ATP binding to its receptor P2RX7, the intracellular influx of K^+^ is responsible for NLRP3 inflammasome activation [4]. Fantuzzi et al. discovered that gene deletion of P2X7R decreased hepatic fat accumulation in mice fed with high-fat diet (HFD), likely because of a blunted hepatic activation of the NLRP3 inflammasome [54]. Fatty acid overload in MASLD induces the derailment of redox reactions and mitochondria dysfunction that leads to an increase in mtROS, thus contributing to the progression of the disease [55]. Additionally, impaired mitophagy represents one of the factors responsible for the progression of MASLD to MASH [4]. In a murine model of MASH, the combination of a high fat, sucrose, and cholesterol diet (HFSCD) and low doses of carbon tetrachloride (CCl_4_) increased the production of ROS and the levels of mitogen-activated kinase protein (MAPK), which in turn triggered the powerful pro-inflammatory signal mediated by NF-kB and NLRP3 activation, leading to further inflammation and fibrosis [56].

### 3.4. Alcohol-Related Liver Diseases (ALDs)

Alcohol-related liver disease (ALD) is one of the main common causes of liver diseases worldwide [57]. The less severe form of ALD is simple alcohol-related steatosis that can progress to alcohol-related steatohepatitis (ASH) and cirrhosis [58]. Alcohol addiction serves as a foundational element in the development of ALD, and NLRP3 appears to play a role in this context as well. Indeed, a study conducted by Li et al. demonstrated that NLRP3-knockout mice exhibit reduced binge alcohol intake and diminished anxiety-like behaviors during alcohol withdrawal compared to those observed in the control mice [59]. Other in vivo studies explored the effects of the IL-1 receptor antagonists in caspase-1, ASC, or type I IL-1β receptor- (IL-1R1) deficient mice, underlining the importance of IL-1β, a product of the NLRP3 inflammasome, in the onset of the inflammation of ALDs [60]. Inflammation and chronic alcohol consumption are strongly connected, since ethanol and its metabolites are the main promoters of hepatocyte necrosis. Upon binding to the P2X7 receptor on the KCs, ATP released from necrotic hepatocytes is responsible for K^+^ efflux and Ca^2+^ influx, which in turn activate the NLRP3 inflammasome.

Along with this phenomenon, ATP enhances hepatic stellate cell activation, extracellular matrix deposition, and pro-fibrogenic processes through the P2X7R-mediated NFLRP3 pathway [57]. Zhang et al. exploited the P2X7R-NFLRP3 pathway as a target for treating alcohol-related liver steatosis, concluding that taxifolin (TAX), or dihydroquercetin, was able to lower the protein levels of cleaved caspase-1, NLRP3, and P2X7R, as well as IL-1β production, in alcohol-treated mice. Thus, they discovered that TAX was able to reduce lipid accumulation and steatosis via P2X7R and NLRP3 suppression in mice with ALD, paving the way for NFLRP3-P2X7R regulation as a potential therapeutic target for alcohol-related liver steatosis [61] (Table 2), although further research is needed.

The activation of the NLRP3 inflammasome in ALD is well documented, and recent findings have identified the involvement of some proteostasis chaperones, the heat shock protein (HSP) 90, and HSP70/HSPA1 in the alcohol-mediated effects on immune cells and liver injury [62]. The innate immune system, including macrophages and the pro-inflammatory molecules network, is impaired in ALD, contributing to the worsening of the disease. Persistent activation of an impaired immune system leads to cell death of the hepatocytes and non-resolving inflammation in patients with ALD [63]. The correlation between NLRP3 and HSP90 within the context of the innate immune response in ALD has been highlighted by Choudhury et al., who found, for the first time, that HSP90 downstreams IL-1β and IL-18 secretion via NLRP3. The blockage of HSP90 inhibits NLRP3, reduces caspase-1 and GSDMD cleavage in macrophages, and prevents IL-1β maturation [64] (Table 2). Therefore, this strategy holds promise as a valuable approach for combating alcohol-related liver disease, although further validation is necessary before drawing definitive conclusions.

### 3.5. Autoimmune Hepatitis (AIH)

Autoimmune Hepatitis (AIH) is an immune-mediated inflammatory liver condition of unclear origin that can manifest with no symptoms, chronic illness, or ALF [65]. Patients with AIH were found to have very high levels of IL-1β that correlated with the aggravation of hepatitis, probably due to the key role of IL-1β as a mediator of inflammation between the macrophages and lymphocytes. In the pathophysiology of AIH, upon antigen presentation, Th0 cells undergo differentiation into T helper type 1 (Th1). Th1 cells secrete IL-2 and interferon γ (IFN-γ) to stimulate CD8+ lymphocytes and activate macrophages in the liver [66,67] (Table 2). Once activated, macrophages secrete IL-1β that aids in the differentiation of Th0 lymphocytes into Th17 cells, whereas impaired Tregs are not able to mediate immunosuppression towards Th17 lymphocytes, further contributing to autoimmunity [25]. Based on these premises, there could be a direct involvement of NLRP3 in the pathogenesis of AIH.

The role of the NLRP3 inflammasome in the pathogenesis of liver damage in patients with AIH is confirmed by the fact that the absence of NLRP3 or caspase-1 ameliorates liver injury and is associated with reduced IL-1β production [66]. A study from Luan et al. showed that IL-1β, NLRP3, and caspase-1 are overexpressed in the liver of Concanavalin A (ConA)-induced AIH mice, while the absence of NLRP3 and caspase-1 reduces hepatocellular damage, thus protecting mice from hepatitis [66]. In the same Con-A murine model of AIH, anti-rhIL1R (an antibody against the IL1 receptor) protects mice from hepatitis by inhibiting the effects of IL-1β, such as immune cell infiltration and the activation of the NLRP3 inflammasome via the elimination of ROS production [66,68].

Another study by Shi et al. revealed a possible off-label effect of dimethyl fumarate (DMF) in the treatment of AIH. Although this drug is used for the treatment of other autoimmune diseases, such as psoriasis and forms of multiple sclerosis (MS), DMF exhibited an inhibitory effect on NLRP3-driven inflammation in a murine model of AIH [68]. Recent studies have discovered that DMF succinylates GSDMD, the executioner of pyroptosis, thus preventing its NLRP3-mediated cleavage and pyroptosis [68,69]. However, Shi et al. also discovered that DMF prevents GSDMD-mediated pyroptosis by inhibiting the upstream mechanisms of GSDM cleavage, including the assembly of ASC complex and the production of caspase-1 [68]. In conclusion, DMF and rhIL1R emerge as promising therapeutic agents for the treatment of NLRP3-driven diseases like AIH and for mitigating NLRP3-mediated inflammation. However, further investigations across different species and under various conditions are urgently needed.

### 3.6. Primary Sclerosing Cholangitis (PSC)

Primary sclerosing cholangitis (PSC) is a rare, idiopathic hepatobiliary chronic condition characterized by inflammation and fibrosis that leads to biliary strictures [29]. The etiology of this disease is not completely understood, but early theories considered LPS from the intestine the major activator of innate immune responses in cholangiocytes, since they are enriched in many isoforms of TLRs. Upon TLR4-LPS binding, biliary cells start producing IL-6, IL-8, and NF-kB, which are fundamental for the pathogenesis of PSC [70]. Genetic variations of the human leukocyte antigen (HLA) are strongly involved in adaptative immune responses, since they determine which gut-derived antigens can be presented to the T cell receptor (TCR) in the CD8+ and CD4+ T cells, supporting the notion of PSC as an autoimmune disease [71,72]. Activated T cells might migrate to the liver, where they contribute to PSC pathogenesis via the promotion of biliary inflammation, which in turn leads to apoptosis and necrosis of the cholangiocytes, along with fibrosis [73].

In human and mice treated with 3,5-Diethoxycarbonyl-1,4-dihydrocollidine for 4 weeks to induce PSC, Maroni et al. discovered that NLRP3 and ASC protein expression is higher in reactive cholangiocytes obtained from diseased subjects, while no detection of these proteins is observed in the healthy samples. In addition, they found that upon LPS + ATP stimulation, cholangiocytes upregulate NLRP3 and increase the production of the IL-6 and IL-18 cytokines, while no effects on NLRP3 activation are observed in terms of IL-1β secretion [74] (Table 2).

Controversially, a study from Gonzalez et al. stated that the NLRP3 inflammasome might be protective towards PSC, since it limits the inflammatory response. The study was conducted in OVAbil mice, representing a model of immune-mediated cholangitis, and OVAbil mice lacking the NLRP3 sensor, and they found that the absence of NLRP3 is accompanied by high levels of inflammation, targeting cholangiocytes. This inflammation was mostly driven by neutrophils, and did not affect IL-1β production [75]. Interestingly, the NLRP3-knockout (KO) murine model used by Maroni et al. shows a reduction in the bile duct mass in mice with PSC, along with a lowering of collagen deposition [74]. The explanation for why these two studies provide different results might reside in the different types of murine models that were used, and for this reason, further studies are necessary to deeply investigate the role of the NLRP3 inflammasome in PSC.

Currently, patients are typically administered UDCA at dosages ranging from 15 to 20 mg/kg/day. However, for individuals who do not exhibit a favorable response to ursodeoxycholic acid (UDCA), exploration of novel therapeutic options is underway. These options include nurhucolic acid (NCA), farnesoid X receptor (FXR) agonists, and peroxisome proliferator-activated receptors (PPARs). In cases where pruritus is present and bile duct strictures are identified, endoscopic treatment may be considered. This may involve dilation, with or without stenting, using a balloon. Pharmacological alternatives include bezafibrate or rifampicin [76]. Acknowledging the pivotal role of NLRP3 in the pathogenesis of PSC, and considering the limited success of individual drugs or therapies in improving transplant-free survival rates in PSC patients, investigations into the involvement of the NLRP3 inflammasome in PSC are still at an early stages of development [72].

### 3.7. Primary Biliary Cholangitis (PBC)

Both primary sclerosing cholangitis (PSC) and primary biliary cholangitis (PBC) are members of the same immune-mediated cholangiopathies family [29]. The difference between these two diseases is due to the anatomical area of interest, since PSC is characterized by injury of medium to large extra- and intra-hepatic bile ducts, while PBC mostly affects the small intrahepatic ducts [77]. In both diseases, one of the mechanisms related to innate immune response and inflammation is the NLRP3 inflammasome. In the case of PBC, the connection between galectin-3 (Gal-3) and NLRP3 leads to the development of the Th17 immune response, resulting in cholangiopathies and fibrosis.

To confirm the role of the Gal-3-NLRP3 axis in PBC, Arsenijevic et al. induced PBC in mice by Novosphingobium aromaticivorans infection, and they found that Gal-3^+/+^ mice had higher levels of NLRP3-expressing dendric cells and macrophages, higher protein expression of NLRP3 and ASC, and a prominent production of IL-1β when compared to Gal-3^−/−^ mice [78]. Without any correlation with Gal-3, studies conducted in patients with PBC demonstrated that mRNA and protein levels of NLRP3, NLRP1 inflammasome, caspase-1, and IL-1β were abnormally upregulated when compared to those of healthy subjects [79,80] (Table 2).

To support the role of NLRP3 in the pathogenesis of PBC, Frissen et al. proved the beneficial effect of NLRP3 deficiency in a bile duct ligation (BDL) murine model. They found that NLRP3-deficient mice possess fewer and smaller bile infarcts, a lower infiltration of CD11b+ Ly6G+ cells, and lower mRNA levels of TNFα, IL-6, and IL-1β with respect to wild type (WT) mice undergoing BDL [80]. However, further studies are needed to elucidate the mechanism of Gal-3 in different stages of PBC diseases and its relationship with the NLRP3 inflammasome. Currently, additional in vivo and in vitro studies regarding NLRP3 blockage using the MCC950 pharmaceutical treatment are needed to determine whether the suppression of NLRP3-mediated signaling might stop the progression of PBC.

PBC patients are currently treated with UDCA at a dosage of 13–15 mg/kg/day. For patients who do not achieve a biochemical response to the initial therapy (ALP > 1.67 × ULN and/or total bilirubin < 2 mg/dL), obeticholic acid is then prescribed. However, it is important to note that the most common adverse effect of obeticholic acid is the worsening of pruritus. For the management of pruritus, we employ a stepwise approach, starting with cholestyramine as a first-line therapy, followed by rifampicin as a second-line therapy. If patients do not respond to these initial treatments, potential alternatives include oral opiate antagonists and selective serotonin reuptake inhibitors (SSRIs) [81].

Approximately 40% of patients with PBC do not exhibit biochemical or clinical responses to conventional therapies. Research has indicated that NLRP3 plays a pivotal role in this context, with Galectin-3 (Gal-3) recognized as an activator of NLRP-3. This suggests that Gal-3 could serve as a potential therapeutic target for PBC treatment, although the precise mechanism of Gal-3-mediated NLRP3 activation requires further elucidation. Currently, ongoing in vivo and in vitro studies on MCC950, a small molecule capable of inhibiting the NLRP3 inflammasome, have demonstrated efficacy in mitigating liver damage in PBC [82]. Such research holds promise for providing therapeutic avenues, particularly for patients who do not respond to UDCA.

**Table 2 ijms-25-04537-t002:** Main studies investigating the involvement of the NLRP3 inflammasome in chronic liver diseases. Abbreviations: HCV: hepatitis C/B virus; HBV: hepatitis B virus; MASLD: metabolic dysfunction-associated steatotic liver disease; MASH: metabolic dysfunction-associated steatohepatitis; ALD: alcohol-related liver disease; AIH: autoimmune hepatitis; PSC: primary sclerosing cholangitis; PBC: primary biliary cholangitis; HCC: hepatocellular carcinoma; IL-1β: interleukin 1β; ASC: apoptosis-associated speck-like protein containing a caspase-recruitment domain; RFAs: rhubarb free anthraquinones; PR2X7: purinoceptor 7; TAX: taxifolin; ConA: concanavalin A; DMF: dimethyl fumarate; Gal-3: galectin-3.

Type of CLD	Effect	Sex and Species	References
HCV	Upregulation of NLRP3 leads increased efflux of K^+^ and IL-1β production in macrophages, but not in monocytes.	Humans	Research paper [37]
	Increased production of caspase-1 results in lipid droplet creation for virus replication.		Review [1]
HBV	NLRP3 inflammasome activation is responsible for enhanced secretion of ASC, IL-18, and IL-1β in hepatocytes.	Humans	Research paper [43]
	NLRP3 triggers humoral immunity against the virus, thus aiding in HBV eradication.		Review article [1]
	Lipid peroxidation, ATP, and impaired mitophagy are the main activators of the NLRP3 inflammasome, triggering the pro-inflammatory cascade.		Review article [4]
MASLD and MASH	MASH patients have increased sera levels of caspase-1, while mice exhibited high hepatic mRNA levels of caspase-1 and IL-1β sera levels.	Male, C57BL/6 mice	Research paper, Review article [60,61]
	RFAs directly inhibit NLRP3 in vitro and improve MASLD in mice.	Male, C57BL/6 mice	Research paper [53]
	Pr2X7 deletion decreases hepatic fat accumulation in mice, through the blunting of NLRP3 activation.	Male, Pr2x7^−/−^ mice and WT C57BL/6 mice	Research paper [54]
ALD	TAX is able to ameliorate alcohol-related liver steatosis via P2X7R and NLRP3 suppression in mice.	Male, C57BL/6 mice	Review article [61]
	HSP90 downstreams IL-1β and IL-18 secretion via NLRP3. Targeting HSP90 reduces caspase-1 and GSDMD cleavage in macrophages.	Female, C57BL/6 mice	Review article [64]
AIH	IL-1β, NLRP3, and caspase-1 are overexpressed in the livers of ConA-induced AIH mice. The inhibition of the IL-1 receptor decreases ROS production and NLRP3 inflammasome activation in mice.	BALB/c mice, Female, C57BL/6 mice	Research paper, Research paper [75,77]
	DMF inhibits NLRP3-driven inflammation in mice with AIH.	Female, C57BL/6 mice	Research paper [68]
PSC	Reactive cholangiocytes from mice with PSC present higher levels of NLRP3 and ASC.		Review article [74]
	A lack of NLRP3 leads to inflammation, targeting cholangiocytes in OVAbil mice.	Male, C57BL/6 mice	Research paper [75]
	NLRP3 and ASC levels are abundant in Gal-3^+/+^ mice with PBC.	Female, Lgals3^−/−^ mice and WT C57BL/6 mice	Research paper [78]
PBC	Patients showed high levels of NLRP3, NLRP1 inflammasome, caspase-1, and IL-1β.	Humans	Research paper, Research paper [83,84]
	NLRP3 deficiency leads to an amelioration of PBC in mice.	Nlrp3^−/−^ mice and WT C57BL/6 mice	Review article [80]

## 4. The Role of NLRP3 Inflammasome in the Post-Liver Transplant Setting

### 4.1. Liver Ischemia-Reperfusion Injury (LIRI)

Liver ischemia–reperfusion injury (LIRI) is a severe development that can occur after surgical procedures, such as hepatic resection and LT, causing several degrees of hepatic dysfunction and ultimately progressing to liver failure. LIRI is typically divided into two phases: ischemia, which is characterized by cell injury due to hypoxia, and reperfusion. During reperfusion, inflammation occurs after blood flow restoration [85]. Currently, no therapeutic strategies are available, and understanding the pathogenesis of liver damage is key for paving the way to new treatment strategies.

Macrophages have been recognized as the main cells responsible for NLRP3 activation via ROS production during LIRI [70,86]. In fact, the knockdown of NLRP3 in mice leads to a reduction of serum alanine aminotransferase levels and a decreased production of pro-inflammatory cytokines such as IL-1β, IL-18, IL-6, and TNFα, leading to liver protection from ischemia–reperfusion damage [87].

The activation of NLRP3 during LIRI is modulated by different mechanisms, including the heat shock factor 1 (HSF1)-beta-catenin axis that activates NLRP3 through the regulation of X-box binding protein 1 (XBP1) [88] (Table 3). Another mechanism includes the release of ROS from damaged cells that stimulates the thioredoxin-interacting protein (TXNIP), which in turn promotes the activation of NLRP3 [89]. This pattern could be modulated by the use of hypothermic oxygenated perfusion (HOPE), employed before LT to reduce the injury of the graft and avoid the risk of primary nonfunction (PNF) [90] (Table 3).

Extracellular histones have been recognized as novel DAMPs that activate the NLRP3 inflammasome in KCs through the TLR9-dependent generation of ROS. This mechanism, in turn, exacerbates hepatic damage by recruiting neutrophils and inflammatory monocytes [83]. Additionally, HMGB1 is another DAMP identified as an LIRI inducer [84]. The impact of the NLRP3 inflammasome on the pathogenesis of LIRI has been highlighted by several studies demonstrating that the lack of NLRP3 or caspase-1, or the blockage of IL-1β, improves disease pathology [83,84]. By inhibiting NF-kB and NLRP3 mRNA expression, silibinin, the principal constituent derived from milk thistle, can reduce inflammatory infiltration, hepatocyte degeneration, and endothelial injury during ischemia-reperfusion damage [91]. A study from Song et al., for example, found that Bruton’s tyrosine kinase (BTK) inhibitor, ibrutinib, is able to reduce ischemia–reperfusion injury by suppressing the activation of the NLRP3 inflammasome in KCs [92] (Table 3).

Isoflurane, commonly used as an inhaled anesthesia, has been shown to reduce the LPS-induced NLRP3 activation in murine macrophages. Its usage as graft pre-treatment in a murine model of IRI reduces the effect of ischemia–reperfusion injury, thus becoming a potential therapy for the prevention of LIRI [93]. The use of pharmacological blockers targeting the NLRP3 inflammasome in liver ischemia during static cold storage or through extracorporeal organ support could be a suitable strategy to increase the success of liver transplantation.

### 4.2. Acute Rejection

Acute rejection is one of the most frequent complication after LT; up to one-third of patients experience at least one episode of acute rejection [86]. Cytokines play a fundamental role in the development of acute rejection. An upregulation of NLRP3-dependent pro-inflammatory elements such as IL-1β, TNF-α, and IL-6 is typical during graft rejection [94] (Table 3). At the beginning of the inflammatory process, ATP and mitochondrial DNA (mtDNA) are important activators of the NLRP3 inflammasome during acute rejection via the NF-kB pathway [95,96]. Notably, all transplant recipients exhibit higher levels of circulating mtDNA compared to those of healthy controls for the first days after LT, even though patients with early graft dysfunction still have elevate levels of mtDNA [97] (Table 3).

Once activated, the NLRP3 inflammasome triggers caspase-1 activation and IL-1β secretion [98]. An inhibitor of caspase-1, Ac-YVAD-CMK, reduces the production of IL-1β in a rat grafts model, leading to anti-apoptotic, anti-inflammatory, and neuroprotective effects [99] (Table 3). Another promising molecule is MCC950, an NLRP3/Caspase-1/IL-1β inhibitor previously suggested for the treatment of MASLD, which improves graft function after LT in pigs [100]. All these strategies show the potential for enhancing the long-term outcomes after LT. Therefore, efforts to gain a deeper understanding of their mechanisms and to validate their efficacy should be prioritized.

**Table 3 ijms-25-04537-t003:** Main studies investigating the involvement of the NLRP3 inflammasome in the post-liver transplant setting. Abbreviations: LIRI: liver-ischemia–reperfusion injury; HSF1: heat shock factor 1; XBP1: X-box binding protein 1; BTK: Bruton’s tyrosine kinase; IR: ischemia–reperfusion; Ac-YVAD-CMK: acetyl–tyrosyl-valyl-alanyl-aspartyl–chloromethyl ketone.

Post-Liver Transplant Setting	Effect	Sex and Species	References
LIRI	NLRP3 activation is modulated by HSF1 and XBP1. It might also be activated by ROS, which in turn stimulates TXNIP.	HSF1^M-KO^ mice	Research paper, Review article [96,97]
	Ibrutinib, a BTK inhibitor, is able to reduce IR injury by suppressing NLRP3 activation in Kupffer cells.	Male, C57BL/6 mice	Research paper [92]
Acute Rejection	Transplant recipients have higher levels of mtROS, an NLRP3 activator, compared to healthy subjects.	Humans	Research paper [97]
	Ac-YVAD-CMK, a caspase-1 inhibitor, reduces the production of IL-1β in rat graft models.		Review article [99]

## 5. Concluding Remarks

Acute and chronic liver diseases are characterized by hepatocyte injury and a significant inflammatory milieu [10]. Among the amplifiers of the pro-inflammatory signals, is the NLRP3 inflammasome, which is the major contributor to the cytokine cascade [6]. Regardless of the classification of liver diseases as acute or chronic, NLRP3 is a key player in the deterioration of the pathology. Many promising in vivo studies on mice have employed antioxidants as therapeutic targets for DILI, alcohol-associated hepatitis, MASLD, and MASH [23,24]. Further studies regarding the role of antioxidants in the regulation of the NLRP3 inflammasome are required to determine whether they might also be used in the management of cholestatic disorders. Although the knowledge regarding NLRP3 inflammasome activation in liver diseases is still very general, some studies on the role of this protein in HBV, HCV, and PSC are still controversial. For this reason, a better understanding of its participation in the development of liver injury is needed to achieve concordance and pave the way for the discovery of new potential therapeutic targets.

## Figures and Tables

**Figure 1 ijms-25-04537-f001:**
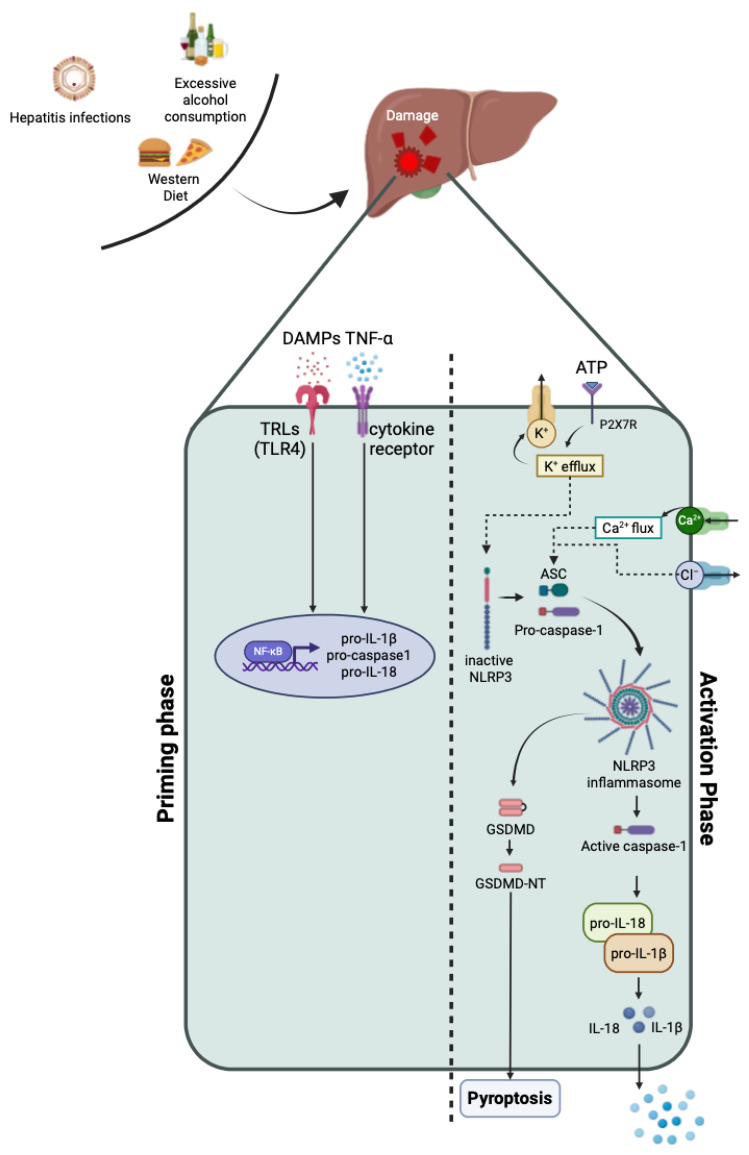
Graphic representation of the NLRP3 priming and activation phase. In the priming phase, upon DAMPs and TNF stimulation, the NF-kB pathway is activated and promotes the gene transcription of NLRP3 inflammasome components such as pro-IL-1β, pro-IL-18, and pro-caspase-1. During the activation phase, once ATP bonds to its receptor, the opening of the K^+^ channel leads to an outflow of K^+^, activating the NLRP3 inflammasome. Other stimuli are responsible for NLRP3 inflammasome assemblage, including high extracellular levels of Cl^−^ and Ca^2+^ flux. At this point, the NLRP3 recruit ASC and pro-caspase-1, leading to the cleavage of pro-caspase-1 into caspase-1, which in turn cleaves pro-IL-18 and pro-IL-1β into their mature forms. In some cases, caspase autoactivation causes the cleavage of GSDMD and pyroptosis, also known as cell death. Figure components were created using BioRender accessed on 8 January 2024 (BioRender.com). Abbreviations: DAMPs: damage-associated molecular patterns; TNF-α: tumor necrosis factor- α; TLRs: Toll-like receptors; ATP: adenosine triphosphate; P2X7R: purinergic P2X7 receptor; K: potassium; Ca: calcium; Cl: chlorum; ASC: apoptosis-associated speck-like protein; NLRP3: NOD-, LRR-, and pyrin domain-containing protein 3; IL: interleukin; GSDMD: gasdermin-D; GSMD-NT: gasdermin-D N-terminal domain.

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
