# Peer review of "NLRP3 Inflammasome in Acute and Chronic Liver Diseases"

_ijms, 2024, doi:10.3390/ijms25084537_

Round 1

Reviewer 1 Report

Comments and Suggestions for Authors

This review assesses an intracellular complex,NLRP3, in acute and chronic liver diseases. In addition how  post liver transplantation complications might be orchestrated by NLRP3 at a molecular level is discussed. The spectrum of drug induced liver injury, alcoholic liver disease and hepatic inflammation in general is thoroughly reviewed. Specifically chronic viral C and B hepatitis is embraced. Considering the most current cause of hepatitic disease,steatohepatitis(both alcoholic and non- alcoholic) worldwide, what have you found in your studies in Padua? Specifically in alcoholic hepatitis with a goal of stopping the ongoing liver injury as well as curtailing the addiction is NLRP3 a promising target or mediator? What is your(Padua) experience with the PSC and PBC patients with and without pruritus?

Reviewer 2 Report

Comments and Suggestions for Authors

Thank you for the opportunity to review the manuscript by Sayaf et al.

Therapeutic implications of NLPR3 in DILI. Therapeutic implications of treating the NLPR3 in DILI may be challenging. Clinically, DILI presents with transaminitis indicating that liver injury has already happened.  Therefore, the offending drug is removed or withheld followed by acetylcysteine to restore hepatic glutathione. As outlined in 2.1, GSH depletion ultimately triggers NLRP3.

Section 2.2. The authors’ objective was to discuss the role of NLRP3 in acute / chronic liver disease. As outlined in DILI, the current clinical approach directly targets redox reactions upstream of NLRP3. This section reviews examples of approaches which target reactive species but falls short of a thorough review of NLRP3 implications in alcohol-induced liver injury.

Section 3.1. A review article should be focused on coverage, analysis / discussion, relevance, and potential impact. Given the effectiveness of directing-acting antivirals in treating and eliminating HCV, a suggesting to target NLRP3 to reduce HCV-infected cell death is antiquated. Even after reaching a sustained virologic response, patients with HCV infection continue to struggle with chronic liver disease and the risk of developing hepatocellular carcinoma or progression to end-stage liver disease. A discussion of NLRP3 in that context would be much more relevant and impactful. This also applies to section 3.2.

Section 3.3. The authors are highly encouraged to use the updated terminology for steatotic liver disease and metabolic-associated steatohepatitis. Otherwise, section 3.3 provides an excellent outline for content to apply to revisions in the other disease-focused sections. Sections 3.5 – 3.7 are also well structured. The authors should ensure the content and depth provided in these sections is consisted across the manuscript.

Section 3.4. Even in the case of ALD and the TAX drug, the effects are likely mediated by the antioxidant capacity of the drug that ultimately suppressed some downstream aspects of the NLRP3. More specifically, NFLRP3-P2X7R modulation is mediated by the function of the antioxidant and not directly targeted. Similar to the previous example, my concern is that the authors suggest NFLRP3-P2X7R may be a therapeutic target but only based on evidence of its therapeutic modulation through an upstream target.  After reading the section, the takeaway is that redox regulation leading to a potential therapeutic benefit is accompanied by alterations in NLRP3 signaling. I believe this satisfies the reviews objective of describing the role of NLRP3 in acute and chronic disease, but the implications for therapeutically targeting NLRP3 components is confusing given the reactive metabolites leading to the pathology would remain unaddressed.

Comments on the Quality of English Language

Comprehensive revisions for English language are required. Most issues are grammatical and throughout the manuscript (repeated use of “and”, “by being”, “thus resulting to be”, incorrect tense, assumption in lieu of consumption, “to further substain”, “very various”, “main common causes”, word order issues, and many other examples).  There are also several places (ex. not autoimmune disease, a Chinese study, not so difficult to imagine, and many other examples) where the language is a bit informal for a scientific manuscript and revision is recommended.

The first paragraph of the introduction provides an excellent introduction.  However, the paragraph needs to be formatted into smaller paragraphs. The individual lines of Figure 1 legend provide an excellent outline for formatting the introduction into paragraphs. There are similar issues with block multipage paragraphs in several other sections (2.1).

Reviewer 3 Report

Comments and Suggestions for Authors The article by Sayaf and collaborators is very well written, easy to read and updated on a topic that should provide more studies in the future, and is fully advisable for publication in your magazine.

Author Response

We thank the reviewer for the comment.

Round 2

Reviewer 2 Report

Comments and Suggestions for Authors

The authors' have addressed my concerns.

Author Response

We thank the reviewer.